# Specific Neural Mechanisms of Self-Cognition and the Application of Brainprint Recognition

**DOI:** 10.3390/biology12030486

**Published:** 2023-03-22

**Authors:** Rongkai Zhang, Ying Zeng, Li Tong, Bin Yan

**Affiliations:** Henan Key Laboratory of Imaging and Intelligent Processing, PLA Strategic Support Force Information Engineering University, Zhengzhou 450001, China

**Keywords:** self-awareness, uniqueness, fMRI, EEG, behavioral

## Abstract

**Simple Summary:**

Self-cognition is unique to the human brain. The brain provides an advantage in prioritizing self-information, which has been found in ethology and brain imaging. Self-advantages are presented as fast response, high attention and strong memory. The specifics of self-cognition can be applied to brainprint recognition. Brainprints analyze the brain response of users when watching identity information sequences, which is considered to be a valuable exploration of intrinsic identity authentication.

**Abstract:**

The important identity attribute of self-information presents unique cognitive processing advantages in psychological experiments and has become a research hotspot in psychology and brain science. The unique processing mode of own information has been widely verified in visual and auditory experiments, which is a unique neural processing method for own name, face, voice and other information. In the study of individual behavior, the behavioral uniqueness of self-information is reflected in the faster response of the human brain to self-information, the higher attention to self-information, and the stronger memory level of self-reference. Brain imaging studies have also presented the uniqueness of self-cognition in the brain. EEG studies have shown that self-information induces significant P300 components. fMRI and PET results show that the differences in self and non-self working patterns were located in the frontal and parietal lobes. In addition, this paper combines the self-uniqueness theory and brain-print recognition technology to explore the application of self-information in experimental design, channel combination strategy and identity feature selection of brainprints.

## 1. Introduction

The research on self-cognition [1] has always been a hot field in psychology, cognition and philosophy. Self is the cognitive basis [2] of the individual to the world, and a large number of researchers have conducted in-depth study of the self-problem. As an important identity attribute in social environment and personal life, self-information processing has unique cognitive processing advantages in psychological experiments and has become a research hotspot in psychology and brain science. The internal self also has external objective embodiment, which includes the self-semantic concept and cross-modal information [3]. For example, self-face [4] is the most intuitive self-related representation and an important symbol of personal identity and self-concept [5]. As an abstract form of self-expression, a person’ s name represents identity, honor and status in social activities [6]. The advantage of self also extends to the body parts [7] such as hands and feet [8]. Researchers [9] have found that the important position of self is also reflected in the effect of self-advantage, which is manifested in the uniqueness of self-information processing. Studies [10] have found that self-prioritization is not only effective in vision and hearing, but also extends to different senses such as taste and touch. By summarizing the research on self-information processing, we find that its advantages can be summarized into three main aspects: the priority of self-identification [11], the sensitivity of self-information [12] and the memory advantage of self-reference [13].

The individual’s self-cognitive advantages are not only reflected in behavioral data, but also in unique in brain imaging results. EEG and functional magnetic resonance have become important technical means in self-cognition research. The temporal resolution advantage of EEG focuses on the components of neuronal discharge. The spatial resolution advantages of functional magnetic resonance and PET provide important brain regions for self-processing. Studies have shown that individuals induced specific early and late EEG components [14,15] in the process of self-recognition, and unique functional area activation [16] appeared in the cerebral cortex.

EEG and functional magnetic resonance imaging (fMRI) have become important technical means in self-cognition research. EEG has a temporal resolution advantage, which focuses on the components of neuronal discharge, while fMRI and PET have a spatial resolution advantage, which provides important brain localization for self-processing. Studies have shown that specific early and late EEG components have been induced in the process of self-recognition, and unique functional activation appeared in the cerebral cortex. With the continuous development of brain imaging technology, researchers have widely used observation methods, such as electroencephalogram (EEG), functional magnetic resonance imaging (fMRI), and positron emission tomography (PET) in experiments, providing research data on brain mechanisms such as nerve discharge and cortical activation for self-cognition. The self-cognition research based on EEG focuses on potential changes and oscillation components [17]. The advantage of the high temporal resolution of EEG is that it can observe the millisecond-level neural electrical signals and record the neuronal discharge changes of the brain in the recognition, coding and cognitive stages of self-information. Studies have found that self-information was unique in the early and late stages of cognition. The N170 and N250 components in the early stage of cognition showed the coding specificity of self-face, while the amplitude and latency of P300 potential in the late stage of cognition were the most significant components of the difference between self- and non-self-information. The spatial high-resolution advantages of fMRI and PET provide the localization of brain functional areas in cognitive processing and record the brain response through the changes in blood oxygen saturation and material metabolism [18]. Studies have identified important brain functional areas such as the frontal lobe [19], cortical midline structure [20] and temporal lobe [21] of self-processing, and extensively discussed the brain skewness of self-processing. However, a single brain imaging method is insufficient to observe the details of cognitive processes, which requires a combination of the temporal resolution of EEG and the spatial resolution of fMRI. At present, there are studies using multimodal brain imaging methods to jointly analyze the self-cognitive process [22]. The above-mentioned self-uniqueness in brain imaging inspires real-world brainprint recognition, such as identification features, electrode combination strategies, device selection, etc.

The uniqueness of the brain is the basis of identity authentication. A large number of identity authentication studies choose the unique processing of self-information as a breakthrough and awaken the special working mode of the brain by presenting the subjects with their own attribute information. This paper summarizes the research on the uniqueness of self-processing and summarizes and combs the application of self-cognition in the field of identity authentication. This paper summarizes the characteristics of self-information processing from multiple perspectives, combs the theoretical hypotheses of different self-cognition mechanisms and analyzes the brain mechanism of self-cognition advantages by combining various experimental methods. The above summary provides a research reference for further exploration of the uniqueness of self-information processing, and provides a theoretical basis and prior information for identity authentication based on ‘brainprint’, especially in paradigm design and identity feature extraction.

## 2. Behavioral Uniqueness of Self-Awareness

A number of studies have shown that individuals pay more attention [23] to stimuli that contain self-information, and the brain has a specific response to self-information [24]. Individual advantages can be summarized as priority [25], sensitivity [21] and long-term memory [26]. The advantages provide a reference for the selection of experimental materials and data processing in identity authentication. First, the priority of self-recognition is mainly manifested in the speed advantage. Self-related stimuli are self-prioritized in the three attention systems: vigilance, orientation and executive control [27]. Individuals’ behavior and neural responses to their own information are significantly faster than information of others [28]. Researchers observed the speed advantage [29] of self in face and name experiments, and the observation data of behavioral keys and neural reactions verified the priority of self-information processing [30]. Secondly, the individual is more sensitive and alert to self-information, mainly when the brain is in an unconscious or distracted state, during which it can automatically process self-information. Researchers [31] found that the perception threshold of self-information was lower. Subliminal experiments [32] showed that self-information mad it difficult to produce ’attentional blink’, and self-related stimuli were harder to mask. The distraction effect of self as a distractor is obvious in distraction tasks. Researchers found that the brain automatically captures self, and self-information distractors are allocated more attention resources. Finally, self-reference processing can effectively improve the memory level, and memory materials are better coded after being associated with themselves. The self-reference effect (SRE) shows that identity authentication achieve a better recall rate and recall effect when memory materials contacted with themselves. Stimulating materials rely on rich self-experience to form a reliable memory structure [33]. Self-reference provides more memory pathways and memory traces for stimulating materials [34]. Individual behavior is realized as the external body of brain instructions, and the relevant conclusions are consistent with and mutually corroborating the results of brain imaging.

### 2.1. Priority of Self-Awareness

The earliest self-cognition experiment uncovered the behavior priority of own information processing; the experiment showed that subjects recognized their own facial images or names faster. The initial experimental records showed the behavior of pressing keys and eye movements, and further studies also found the priority recognition of the subjects’ own information in the neural response signal. This work summarizes the literature on self-awareness as shown in Table 1.

In behavioral studies, multiple experiments repeatedly verified the processing priority of subjects’ own faces and names. Self and non-self face recognition as the most intuitive stimuli has been widely studied. Tong [11] found that self-face recognition has a stable speed advantage. Even after the participants were trained to watch a large number of other face stimuli, the processing speed of their own faces was still faster than that of strangers’ faces. In the multi-angle face recognition test, the search speed of the front, side and inverted faces between the subjects own selves and strangers was compared. The subjects always searched their own faces faster than the strangers. In an additional attention study, the experiment found that self-target recognition in the state of concentration and distraction showed a speed advantage. Keenan [42] added a control stimulus acquaintance in his study, and the familiarity with colleagues was between himself and strangers. The experiment tested the speed and accuracy of keystrokes for participants to identify their own faces and other faces. The results showed that when subjects use left-handed keys, the recognition speed of their own faces is faster than that of familiar faces and strangers, and they show a speed advantage for self faces in both forward faces and reverse faces. The following research draws on the reference of familiar faces; Sui [36] found that the average reaction time of self-faces was 551 ms, which was significantly faster than the recognition time of familiar faces (596 ms) and unfamiliar faces (588 ms), while there was no statistical difference in the reaction time of familiar faces and unfamiliar faces. Jie [37] found that independent individuals showed unique processing of self-positive faces, and their own faces showed higher recognition accuracy and faster search speeds than familiar and unfamiliar faces. Through functional magnetic resonance imaging data, it was found that the activation and rapid response of the right frontal lobe were related to self-face recognition. In addition, the experiment of irrelevant face recognition also has self-cognitive advantages. Ma [38] takes face orientation judgment as the experimental task, and the identity attribute of face is the irrelevant task. The experiment found that the key-press speed of self-face orientation in the non-threatening state was faster than that of acquaintances and disordered faces, and the results showed that self-faces in both explicit and implicit experiments had fast recognition advantages.

As a highly abstract social identity, the rapid identification of names has become a research hotspot [6], and names have gradually become a common stimulus in self-cognition [43]. In the self-name recognition experiment, Harris [39] studied the self-name fast search problem through nine visual search experiments. The names of 60 subjects were detected faster than other identity names. The author believed that subjects’ own names are a ‘high priority’ text stimulus.

Researchers have used eye tracking technology in the task of capturing self-visual information, recorded the eye movement data of subjects in task search and analyzed the temporal and spatial attribute information such as fixation time, fixation point position and eye hops. Wang [44] found that the average number of eye beats for subjects searching for their own names was 1.1 times less than that of their mothers and celebrities, and the first fixation time of capturing their names was 170 ms faster than that of other names. The experiment found that subjects’ own names had stable and obvious search advantages, and the results showed that the eye movement index and search behavior of individuals were more efficient in when they searched their own names. In the recording of neural responses, Tacikowski [41] recorded the EEG signals of subjects with different names; results showed that compared with the names of celebrities and strangers, the latency of P300 induced by their own names was shorter.

### 2.2. Sensitivity of Self-Information

Individuals are naturally sensitive to their own related information, which is mainly manifested in that their own information is more likely to awaken the brain in unconscious or distracted states, and their own related stimuli are harder to mask by shortening the presentation time and adding a noise figure. Using subjects’ own characteristics as distractors has a stronger interference with the participants. This work summarizes the literature on self-sensitivity as shown in Table 2.

In the self-recognition of non-attentional states, the cocktail effect [12] is the most famous. In the experiment, participants have a certain probability of noticing their own names in the non-follower’s ears, that is, the brain can capture self-information in an unconscious state, indicating that individuals can still process their own names in a distracted state. Several scholars repeated a similar cocktail party experiment and evolved a variety of unconscious attention paradigms about their own information, verifying the brain’s sensitivity to self-information under multiple sensory stimuli. Alexopoulos [45] believes that the familiarity of others is the influencing factor of face recognition. In order to explore the attention advantage of self-information in detail, four attention experiments of subjects’ own names and other names were carried out. By comparing the individual’s response time to self and others’ information in distraction tasks, it was found that self-names have a stable attention advantage, while acquaintance names have little hint effect. Due to the lower possibility that a short presentation duration of a stimuli can be consciously modulated, attention capture of self-information is considered unconscious and automatic, and this automatic processing can promote information storage and integration. Researchers not only found unconscious processing of self-information in auditory distraction tasks, but also verified unconscious self-cognition in visual name tasks. Callan [46] explored the brain response to passive sound stimulation in fMRI experiments and found that voice processing involves self-monitoring functional areas. The increased use of auditory–motor self-monitoring leads to different activation of auditory–motor-processing-related brain regions in singing and speech. Pfister [47] designed the subliminal priming experiment of noun classification, and the results showed that self-names would attract attention and be prioritized when the subjects were unconscious. Studies have found that self-names can also affect individual behavior in unconscious states. The authors propose that the brain seems to have an automatic recognition and processing mechanism for self-information.

Self-information is more difficult to mask under the threshold, and the brain needs less attention resources for self-information cognition. Shapiro [47] tested the ‘attentional blink’ phenomenon of different nouns, including subjects’ rapidly changing names of themselves, others and ordinary nouns. The results showed that the fast-rendering method makes it difficult for the brain to produce attentional blink to subjects’ own names stimulation, and subjects’ own names show stronger anti-interference ability. Shelley [48] studied the brain attention allocation for low-level and pre-attention and explored the role of subjects’ own names in attention capture in masking tasks. The experiment carried out positive and negative tests on the attention capture ability of subjects’ own names. In the first stage, the probability of subjects’ own names and other symbols being masked as tar-gets was tested. The results showed that the probability of the subjects’ own names being identified in the mask task was higher. In the second stage, the self-name was used as a mask to test the ability to interfere with the target. The data show that the probability of self-name masked symbols being identified is lower. The author believes that the distribution of attention is strongly influenced by the information in conscious and unconscious states. Pannese [49] instructed subjects to match gender in very short face presentation durations and provided evidence of self-prioritization through subliminal priming experiments. The experiment set up the target faces of oneself, acquaintances, celebrities and strangers. The results show that when the face presentation time is 17 ms, there are only differences in response time between the subjects’ own faces and gender in matching and mismatched cases, which verifies that self-information can also be captured and processed in a very short time. The authors believe that self-information has obtained special cognitive processing, and self-faces benefit from early brain processing. Researchers [50] conducted a quantitative analysis of cognitive load in attentional blink tasks to guide subjects to judge the gender of their own, celebrity and stranger names under the threshold. The results show that the gender judgment accuracy of the subjects’ own names under low-load and short-delay conditions is significantly better than that of other names, but the difference in the discrimination of different names under high-load conditions disappears, indicating that self-uniqueness is still limited by attention re-sources.

The interference effect of self-information as a distractor on main tasks is more obvious. Researchers [51] conducted two sets of self-face attention control experiments. One group of subjects used their own names as target stimuli and friends’ faces as interference objects, and the other group used friends’ names as target stimuli and their own faces as interference objects. The results show that the interference effect of the subjects’ own faces is much stronger than that of their friends. The author speculates that the subjects’ emotional value and familiarity improve the attractiveness of their own faces, which makes their faces particularly difficult to be ignored. When the interference object is one’s own face, the individual’ s attention will be captured by their own face, resulting in a stronger inhibitory ability of self-face to target stimuli. Devue [52] conducted an in-depth analysis of the spatial position of his own face as an interference, and the experiment placed his own interference inside and outside the focus of attention. The results show that self-faces can temporarily attract attention, but distractions need to appear in the focus of attention. In a subsequent study, Devue [53] used an eye tracker to record the behavioral response of his own face as an interference, in order to verify whether the attentional location of the face originated is from priority processing or visual retention. In the experiment, different mouth types were designed to identify the subjects. The results showed that the reaction time of the face as a distraction was longer. Eye movement data showed that the subjects only stayed on their own faces for a longer time, which proved that their own faces were more difficult to disengage. In addition to the strong interference of self-face, Yamada [43] found that abstract information about his name also inhibited target stimuli. In the experiment, the visual search task of the target point was designed, and the results showed that when the subjects’ own name appeared as an interference, the average deviation of the eye movement error of the target point positioning was 1.61°. The author believes that the distortion of visual space caused by his name leads to the deviation of spatial distribution of visual attention. Wang [44] believed that when self-information is irrelevant to the central task as a distractor, the brain automatically captures the attentional attribute of self-information. In order to successfully complete the main task of the experiment, participants need to invest more attentional resources to suppress the strong interference of self. Since the distractors of self-attribute cause individual attention dispersion, the brain allocates more resources and attention to unrelated tasks, resulting in a decrease in latency and accuracy of mainline task processing. Wolford [54] conducted an interference test on the subjects’ own names in the non-attention state. The results show that the subjects’ names extend the judgment time of odd and even numbers, and their strong interference affects the implementation of the main task. The author believes that the subjects’ names have lower perceptual thresholds than other words.

Additional studies have found that the audio of the subjects’ own name can cause individual brain responses under a minimally conscious state, such as sleep [55,56], general anesthesia [57,58] and patients with Alzheimer’ s disease [59]. Researchers have found that the subjects’ own names are a very stable and universal stimulus. Self-names not only cause strong brain reactions in awake states, but also produce specific cognition in sleep. Perrin found that the brain will also have a significant response to self-name, for that persist vegetative states, minimally conscious states, atresia syndrome and other micro-conscious patients.

The sensitivity of the subjects’ own information is also widely used in daily life, where medical care is used to detect the basic function of the brain [55]. Accidents and infections during surgical anesthesia can cause severe brain injury, which may cause patients to lose language interaction for several years after surgery [19]. There is an urgent need for a reliable detection of individual consciousness impairment or disability in clinical practice, which is a serious challenge to ensure the patient’s post-operative life experience. The Glasgow Coma Scale is used for coma diagnosis [60], which evaluates brain recovery and deterioration and predicts recovery. However, the evaluation of the scale is difficult to apply to the real-time surgical environment, and its quantitative score depends on personal experience. For example, in the process of first aid, the mental state of coma patients can be judged by calling the patient’s name to test the degree of brain awakening. Auditory responses to the subjects’ names are often used to assess leftover self-consciousness in a state of conscious change (i.e., coma, vegetative state, sleep or sensation). In the rehabilitation test of patients with atresia syndrome and vegetative state, the degree of brain response to their own name and face is an important indicator of the recovery of cognitive function [55].

The brain’s sensitivity to information can be applied to the experimental design of identity authentication, which has important reference significance for experimental material selection and stimulus presentation time. The original identity authentication experiment was stimulated by alphabetic spelling, daily necessities watching and other independent materials. The recent experiment targeted the user’ s personal information as the experimental content and compared it with the brain reactions of other’s information for identity authentication. The accuracy of identity recognition was significantly improved after the introduction of self-related attribute materials. In addition, based on the differences in self and non-self sensitivity, supra-liminal and subliminal experiments can be set. By adjusting the presentation time of stimuli and setting the masking map, self-related stimuli can be presented above the cognitive threshold, rather than self-information below the cognitive threshold. The supraliminal and subliminal cognitive differences expand the difference between self and non-self brain responses. Subliminal experimental stimuli can stimulate an individual’s subconscious and instinctive responses, and become a more hidden and safe means of identity authentication.

### 2.3. Memory Advantage of Self-Reference

Self-reference processing activates the memory advantage of the brain, and combining self-information with experimental materials helps memory. Researchers found that in the memory test experiment, subjects use self-reference strategies to remember the stimulus more firmly—this memory advantage phenomenon is named self-reference effect.

Many studies have summarized the memory advantages of self-reference processing, among which some have proposed that self-reference strategies can obtain better memory scores than other processing and semantic processing [43]. Research [13] found that self-factors play an important role in memory, and the memory effect after stimulation and self-association is stronger than other encoding methods. Jacoby [61] carried out the experiment of self-reference and recall experience probability; the results show that the encoding of characteristic adjectives in self-reference has a more reliable memory effect. In the experiment, participants experienced a memory improvement for the information with low self-correlation and the information with high self-correlation. The authors believe that this is because the priority encoding of self-correlation information is realized by recalling experience and automatic retrieval. In addition to the visual self-reference effect, Greenwald [13] studied the learning and memory of new knowledge in auditory experiments by guiding subjects to combine the target words in sentences with the names of themselves and friends. The results show that the combination of the subjects’ own target words has a significant memory advantage, which were stable and sustained in multiple groups of experiments. By computer-fitting the above results, the author believes that personal experience can automatically play an intermediary role in memory without language expression. Bower [62] expanded self-related memory materials, which not only verified the memory advantage of conventional self-reference tasks, but also found that linking memory materials with personal experience or life fragments could promote memory enhancement. At the same time, the experiment compared the memory effect of self life fragments and strangers’ life fragments, finding that the memory ability of strangers’ memory fragments is poor. The author believes that self-reference and self-event correlation are better memory structures.

Researchers have tried to explain the self-reference effect from various perspectives. Dewhurst [63] studied the problems of memory and self-psychological representation in a subjects with amnesia caused by hypoxia. A personal semantic memory test was designed for patients with brain injury. The author isolated the contribution of plot and semantic memory to the formation of self-knowledge and speculated that the memory advantage of self-reference resulted from the unique memory system of self. Rogers [64] carried out an individual memory experiment on adjectives. Both sub-experiments showed that self-reference tasks performed best in adjective recall. Self-participation leaves complex and complete memory traces in the process of adjective encoding, thus improving the depth of memory of adjectives. It has been concluded that self-reference can promote the brain to encode abundantly and effectively. At the same time, self may be an advanced mode in brain cognition. The self-processing mode is an important system of human information processing and is deeply involved in processing, parsing and memory processes. Symons [65] compared the experimental results of self-related information and used meta-analysis methods to study the basis of memory advantage. Studies have found evidence that self-referenced memory methods are widely used in life. Connecting materials with themselves helps to refine and organize coding information. As a practical memory method, self-reference processes promotes both organizational processing and fine processing.

Self-reference processing and survival processing have memory advantages, and the memory effects of the two types of processing are widely compared. In five experiments, Dewhurst [63] compared the information coding ability of individuals under survival processing and descriptive self-reference tasks and added a moving scene as the control group. The correct rate of word recall showed that individuals had higher recall of self-reference adjectives than survival processing methods. In the experiment of refining adjectives, the advantage of self-reference processing is not affected by the image of adjectives, that is, the memory level of both concrete and abstract adjectives is improved. Experiments [66] showed that the memory effect of self-reference was stronger than that of survival processing, and speculated that social approval judgment in the process of self-reference attracted more attention resources, and higher attention input promoted the improvement in memory.

Individual self-cognitive advantage is not only reflected in behavioral data, but also is unique in brain imaging results. EEG and functional magnetic resonance imaging have become important technical means in self-cognition research. The temporal resolution advantage of EEG focuses on the components of neuronal discharge, and the spatial resolution advantage of fMRI and PET provides important brain localization for self-processing. Research has shown that individuals induced specific early and late EEG components in the process of self-recognition, and demonstrated unique functional activation in the cerebral cortex.

The memory advantage of individual information provides a theoretical support for the security of identity authentication. Self-cognition is the only and long-term memory result. Intruders cannot forge the brain response of real users through short-term memory. Memory advantage is also reflected in functional magnetic resonance imaging, and memory function areas in the hippocampus are activated during self-recognition. Memory advantage is applied in the content of identity authentication experiments, which includes identity information stimuli of self and stranger, such as name words and facial images.

## 3. fMRI and PET Imaging of Self-Information Processing

In fMRI and PET studies, the important brain regions in the self-reference task can be located. Figure 1 shows the cortical regions activated by self-attribute information in several neuroimaging articles. Studies have shown that self-processed cortical structures include the frontal lobe [22], temporal lobe [67], cortical midline [68], and parietal lobe [42]. The location of relevant brain regions has important positioning significance in identity authentication. The lead selection can be carried out according to the scalp coordinates corresponding to the cortex. The corresponding location can be set as the node of interest in the calculation of brain network characteristics. It can also be used as the topological structure characteristics to construct the adjacency matrix of the graph neural network. Tacikowski [41] found that self-cognitive processing requires the cooperation mechanism of multi-brain regions, and the processing of self-reference information involves the mutual cooperation of multiple functions such as memory, recognition, emotion and cognition. A review [60] summarizes the important role of the intraparietal sulcus (IPS) of the supraparietal lobule (SPL), the attention system of the large area cortex and the frontal parietal cortex located at the junction of the parietal lobe, occipital lobe and temporal lobe, which is related to the neural correlation (NCC) of functional mapping and search consciousness.

Studies have generally found that the prefrontal lobe is significantly activated in self-related tasks, and the frontal lobe participates in the neural activities of self-awareness and self-reflection. The region is involved in the distinction between self and others [69] or the attention processing of the subjects’ own faces [70]. In the experiment of Devue [52], participants were instructed to make identity judgment between themselves and their colleagues. The experimental stimuli included real and stretched individual pictures. fMRI data showed that the brain had different activation regions for its own and acquaintances’ faces. During the process of self-recognition, a large area of activation of the right frontal lobe and the priority activation of the right frontal cortex were considered to be the main areas related to visual self-recognition. Kelley [18] studied the memory advantage of self-reference. In the experiment, participants were asked to press keys on themselves, others, cases and related adjectives. The results showed that compared with others and cases, the activation of MPFC during self-cognition was significantly reduced compared with baseline, indicating the selectivity of self-recognition in this brain region. The psychological activity of self-reference belongs to the default network of the brain [71]. Others and events can cause the MPFC to deviate from baseline, which is considered to be interrupted by the default mode of brain function in specific goal-oriented behaviors [72]. The above results of MPFC in fMRI were consistent with those of PET [73]. It can be concluded that the MPFC is related to individual self-reflection and self-reference. Perrin [22] conducted a joint acquisition of EEG and head PET signals when subjects listened to their own names, and found that when subjects heard their own names, they induced significant P3 component. The cerebral blood flow of the MPFC showed a significant linear regression with the P3 component, and the correlation between the P3 amplitude of the subjects’ own names and MPFC was greater than that of any other name, indicating that MPFC played an important role in self-processing. Pathological studies in another direction also provide evidence of the role of the frontal lobe in self-cognition. Researchers tested patients with frontal lobe injuries for self-recognition, and patients showed insufficient self-awareness and self-reflection. Keenan [35] performed self-face recognition on normal subjects after frontal lobe anesthesia. During anesthesia, subjects lost the ability to recognize their own faces, but still could recognize the heads of others. In addition, Ackerly [74] and Damasio observed that frontal lobe injury reduced individual self-recognition ability. The above studies show that the prefrontal cortex, as the brain functional area of advanced cognitive processing, is deeply involved in the self-cognitive process.

The temporal lobe has been widely studied in self-identification. A wide range of cortical areas, including parietal, occipital and temporal areas, are posterior hot areas [19], which play an important role in individuals’ recognition and cognition of the outside world. Even if small areas of the hot cortex are removed, it can lead to the loss of the entire contents of consciousness, such as the inability to recognize faces, colors, and contours. The auditory experiment of Tacikowski [41] compared the brain responses of oneself, important acquaintances, celebrities and strangers’ names, and recorded the behavioral data and cortical activation of the participants by key pressing and fMRI. fMRI results showed that a wide range of bilateral networks, including the frontal lobe, temporal lobe, limbic system and subcortical structures, were activated in the process of self-name recognition compared with names of celebrities and strangers. The authors analyzed the cortical regions of the thalamus, caudate and lentiform nucleus in the subcortical structure, which play an important role in self-name recognition and perform rough and fast information processing. However, when one’s own name is compared to the brain activation map of important acquaintances, there are significant activation differences only in the right inferior frontal gyri (IFG). The significant activation of the right IFG is consistent with the results of previous self-face recognition studies [52,75], which proves that the right IFG plays a key role in self-related information processing. Kaplan [76] also found the uniqueness of the right IFG in self-recognition, and the researchers instructed subjects to identify their own/friend sounds and their own/friend faces, respectively. The results showed that the two sensory tasks had the subjects’ own brain working modes, and only the activation of the right IFG increased in the common tasks of sound recognition and facial recognition. The author believes that the right IFG involves processing a variety of sensory forms of self-related stimuli, which may contribute to abstract self-representation. In a subsequent study, Tacikowski [41] designed the auditory–visual cross-modal pattern in the self-cognitive experiment, and recorded the brain responses of different senses to the subjects’ own information by fMRI. The results showed that the right IFG was the only brain region that showed cross-modal activation of self-face and sound. Because the IFG is located in the ventral pathway of the frontal network, the authors believe that the activation of this region indicates a bottom-up working mode caused by one’s own name. In addition, bilateral IFGs are widely activated in self-cognitive hearing experiments, possibly because attention is automatically shifted to personal names.

With the deepening in self-cognitive model research, Northoff [77] summarized the previous conclusions of self-reference processing, concluding that the cortical midline structure (CMS) is an important functional part of self-reference stimulus processing, and proposed that a CMS-based content management system is a basic component of the self-cognitive model. The CMS regions in this paper included the medial orbital prefrontal cortex, dorsolateral prefrontal cortex, anterior cingulate cortex and posterior cingulate cortex, which correspond to four sub-processes of self-reference stimulation: statement, assessment, monitoring and elicitation. Johnson [68] found that the activation of the anterior medial prefrontal cortex and posterior cingulate cortex was consistent when subjects reflected on their abilities, characteristics and attitudes. The experimental results are consistent with the results of functional brain injury. The involvement of the medial pre-frontal cortex and posterior cingulate cortex contributes to self-reflection. Argembeau [66] studied cross-temporal self-cognitive representation. Participants were asked to think about self and others in the present and past periods. The fMRI results showed that the activation of the prefrontal cortex and posterior cingulate cortex was significantly increased when individuals reflected on themselves. The CMS functional area content management system helps to distinguish between present and past self.

It should be noted that the current brain region positioning conclusions of self-cognition are not consistent, and there are also differences in the degree of cortical activation during self-recognition in different studies. On the one hand, it is due to the different experimental paradigm design. On the other hand, the individual’s cultural and living background leads to differences in self-concept and social identity, and ultimately reshapes the differences in brain structure and function.

## 4. EEG Study of Self-Information Processing

With the advantage of the high temporal resolution of EEG, in the process of identity authentication, time-domain segmentation is carried out according to the EEG components, and corresponding weights are given to the characteristics of different stages according to the component differences of self and non-self signals. The attention of additional differences according to the characteristics of each component helps to improve the classification. The research of self-cognition based on EEG focuses on the self-induced oscillation component, and the experimental paradigm is mainly Rapid Serial Visual Presentation (RSVP). As shown in Figure 2, Zhang [78] compared the EEG responses induced by the names of himself, acquaintances and strangers in his work. The EEG components are divided into pre-cognitive and post-cognitive according to the latency. The pre-cognitive components of self-induced are mainly N170 and N250, and the post-cognitive components are mainly P300.

N170 is the earliest specific component of self-induced. Caharel [79] studied the early EEG components of important facial recognition. By analyzing the N170 components of reaction time and event-related potential, the authors proposed that the emotional ex-pression of self-face was processed before 170 ms, and the N170 component can effectively be used to distinguish familiar and unfamiliar faces. Caharel [80] compared the early EEG data of the faces of self, celebrity and stranger. The results showed that the amplitude of N170 decreased with the decrease in familiarity. The author believes that the N170 component indicates that the coding of the brain on the face is affected by familiarity. In addition, Keyes [81] found that self-face elicited more negative N170 components, suggesting that early brain components were modulated by one’s own facial information. Rossion [82] proposed that N170 is a key event in facial processing. The N170 component in the occipital–temporal region reflects the preferential response of low-level vision in early face recognition. In addition to showing specificity in self-face visual coding, N170 can also be used to distinguish between self and non-self in self-name auditory experiments. Researchers [83] played sentences containing subjects’ own names and common nouns for subjects, both of which appeared at the beginning of the sentence. The study found that the peak amplitude difference between self and non-self appeared at 125–225 ms, and found that subjects lisetning to the sound for about 200 ms under normal speech speed can judge the vocabulary type. There is a debate about the EEG components of early processing of subjects’ own information. The traditional view is that N170 components are not affected by familiarity. For example, in some studies [84,85], there is no difference in N170 components for different identity recognition. Researchers believe that different experimental designs lead to different EEG results.

The pre-cognitive N250 component has also attracted the attention of researchers. Keyes [81] believes that as the most intuitive and unique physical feature, one’s own face is most closely related to self-awareness. N250 is an EEG component related to facial recognition and identity discrimination. N250 in the occipital temporal region can effectively distinguish familiar and unfamiliar individuals. The study of Perrin [86] investigated participants in a sleep state who were played audio stimuli of their own names and other’s names; the results showed that the amplitude of the N2 component induced by subjects’ own names was higher and the incubation period was shorter. Tanaka [87] observed that the N250 component had an obvious amplitude induced by subjects’ own faces in the temporal lobe, and the N250 component had strong robustness in the experiment of distinguishing familiar and unfamiliar faces. The increase in this component was due to the long-term memory of familiar faces. However, N250 also shows strong acquired plasticity, that is, repeated learning of familiar faces can also induce similar N250 components, indicating that this component may not be unique in self-cognition.

The brain visual system showed the specificity of P300 in self-information, and the results were consistent with the visual and auditory experiments. Folmer’s [88] experiment instructed 10 subjects to listen to their own and others’ names. The results showed that the P300 of normal subjects were only induced by their own names, and the author believed that the experiment where subjects heard their own names was helpful to assess the cognitive status of patients with mental disorders. The study of Perrin [22] combined the temporal resolution advantage of EEG and the spatial resolution advantage of PET and found that the P3 component induced by the subjects’ self-name showed a linear relationship with the local cerebral blood flow of the right prefrontal lobe, and the prefrontal high-level cognitive region involved self-judgment and self-reflection. In addition, P3 showed significant regression with the right superior temporal sulcus, and the right temporal sulcus was responsible for identifying personality characteristics and body appearance. The authors believe that the combined analysis of the two brain imaging methods proves that P3 components are deeply involved in self-processing.

Researchers carried out studies on self-processing of patients with specific cognitive impairment. The study of Webb [89] compared the EEG data of normal, autistic and developmentally retardation children in the self-face experiment. The results showed that the EEG delay of autistic children was longer and the response was slower. Compared with the P300 component of face and ordinary objects, the P300 amplitude of autistic children on objects is higher than that of faces, showing the opposite nerve response to normal individuals. The author believes that the abnormal latency and amplitude of autism patients’ EEG data are due to the disorder of brain response modes and verifies the importance of P300 component in face recognition from the side. Signorino [90] performed an oddball paradigm auditory test on coma patients with cortical dysfunction. The results showed that 9 of 16 patients responded to their own name P300, 7 of 9 subjects who were induced P300 survived, while only 3 patients who did not induce P300 survived. The authors’ supplementary study found that P300 was positively correlated with the recovery probability of coma. In addition, Fischer [91] proposed a hybrid assessment method for P300 and mismatch negative (MMN) of coma patients and applied it to 50 patients with severe brain injury who were in a coma for more than 20 days. The authors found that P3 was unique to arousal, and almost all patients with a P300 response to their own name regained consciousness (except one patient with P3b response), indicating that the P300 component reflected a high level of self-cognition in coma patients. Patients with cognitive impairment still show stable self-information processing advantages, and many studies have consistently demonstrated the importance of P300 components.

In addition to studying the event-related potential components of EEG components in the time domain, Mu [92] studied the EEG rhythm changes in self-reference processing, and analyzed the self-processing from low-frequency and high-frequency oscillations. There are differences in the response time, brain regions and synchronization between the low frequency and high frequency of self-cognitive process. The low-frequency component α wave shows event-related synchronization in 400–600 ms of the central region, θ wave shows event-related synchronization in 700–800 ms of the frontal lobe, the high-frequency component β wave shows event-related synchronization in 700–800 ms of the central parietal lobe, and the γ wave shows event-related synchronization in 500–600 ms of the central region. The authors believe that low-frequency components represent the relationship between emotional valence and self-reference. High-frequency oscillation represents the distribution of attention and self-judgment. EEG frequency domain results show that cognitive and emotional processes are separated in the process of self-reference.

## 5. Application of Self-Specificity in Brainprint Recognition

Studies have shown that individuals pay more attention to stimuli containing self-information, and the brain has a specific response to self-information. Individual advantages can be summarized as priority, sensitivity and long-term memory. Advantage characteristics provide reference for experimental material selection and data processing in identity authentication. The specific advantages of self-information processing, such as priority, sensitivity and long-term memory not only have theoretical value, but also have broad application prospects in the emerging biological recognition applications.

With the advancement of sensors and biotechnology, the identity authentication of external biometrics has become mainstream in daily life. However, in the face of more complex and severe identity security attacks, we should not only confirm the external biological characteristics of an individual, but also protect the user’s subjective will in identity authentication. Traditional biometric authentication technology still has some security risks. In fingerprint identification, criminals may synthesize false fingerprints through materials such as gelatin, thus deceiving the authentication system. Some criminals may force legitimate users to complete authentication by force. These security risks put forward higher requirements for biometric-based authentication. How to achieve in vivo detection and stress resistance has become the focus of current research in this field. Therefore, exploring the inherent biometric authentication technology has become an important research topic in the field of identity security. The subjective initiative of the brain matches the cognitive needs of the intrinsic biological characteristics, protects the subjective certification intention of the coerced person, and enriches the reset replacement function of the intrinsic certification content.

Brain signal is a recognition feature of organisms and has individual specificity that can be quickly identified. Extracting the ‘EEG fingerprint’ from EEG signals as a recognition feature has become a new way to authenticate identity. As an inherent individual difference phenomenon, self-uniqueness has become an important theoretical support for the current popular brainprint recognition technology. The effective verification of self-uniqueness in the field of psychology and brain science will support the individual uniqueness theory of brainprint recognition technology, similar to fingerprints. There are considerable individual differences in brain structure and advanced cognitive function, which lays a foundation for the study of brain biological indicators [93].

Brain imaging technology obtains the activity of human brain neurons. At present, the commonly used brainprint recognition equipment is mainly portable EEG equipment, and the EEG equipment collects weak electrical signals on the scalp surface. Compared with other bioelectrical signals, brain imaging technology has unique advantages as a biological feature, mainly reflected in the following aspects: first, EEG signals have strong anti-counterfeiting ability, which is highly dependent on the individual’s brain structure and functional neural activity, and is difficult to replicate and forge; second, compared with traditional biological features such as face, fingerprint and voiceprint, EEG signals have higher concealment, will not be exposed to the outside world, and are not easy to obtain from the far end; third EEG signals have in vivo detection; and finally, EEG signals are stress-resistant, and external stress and mental stress will cause abnormal EEG signals to lead to authentication failure. EEG has more advantages than traditional biometric patterns in anti-spoofing attacks, privacy compliance and liveness detection [93].

The basic principle of EEG-based identity authentication is to find information that can stably represent individuals from EEG signals and build models to achieve accurate classification on this basis. As shown in Figure 3, the basic process includes two parts: registration and certification. In the registration portion, the EEG signals of users and possible intruders are collected first, then, the collected EEG signals are preprocessed to remove the noise introduced by equipment, environment and other factors. On this basis, feature extraction, feature matching, matching score fusion and other operations are performed to generate a specific classification authentication model for each user. The authentication model is stored in the model database to provide data support for the next stage of classification authentication. In the login portion, the tester will first select or enter the login account, and then use the same experimental paradigm (resting state or task state) as the registration phase to complete the authentication test experiment. The system will use the authentication model generated in the registration phase to complete the classification to realize the identity authentication decision. The uniqueness of the individual brain is the basis of identity recognition. At present, the unique neural activity induced by self-information is more obvious and stable. Therefore, researchers have carried out extensive research on the uniqueness of brain self-information processing.

The premise of biometric-based identity authentication methods is to verify the individual uniqueness of the features. By measuring the neuronal activity of the individual brain, combined with the unique self-cognitive attributes of the brain, EEG can be used as an effective means of identity authentication. The experiment presents self-attribute stimulation to induce a specific response of the brain and detects that the brain stimulates a specific response to identify the subject. Identity authentication is reflected by the uniqueness of self, and the unique processing of self-information by the human brain can be used as theoretical support.

The specificity of self-cognition improves the paradigm materials and parameters of brainprint recognition. The sensitivity of the brain to information can be applied to the design of identity authentication experimental paradigms, which is of great significance for the selection of experimental materials and the presentation time of stimuli. The stimulus materials of brainprint recognition have experienced the development of general stimulation, self-stimulation and subliminal self-stimulation. The initial identity authentication experiments stimulated self-independent materials such as letter spelling and daily necessities viewing. The performance of identity authentication is limited by user capacity. When the user capacity is expanded, the performance will drop sharply. The current experiment takes the user’s personal information as the experimental content, and compares the brain response of others‘ information for identity authentication. The accuracy of identity recognition is significantly improved after introducing its own related attribute materials. The memory advantage is applied to the content of identity authentication experiments. The experimental content includes self and stranger identity information stimuli, such as name text, facial images, etc. In addition, based on the sensitivity difference between self and non-self, supraliminal and subliminal experiments can be set up. By adjusting the presentation time of stimuli and setting masking maps, self-related stimuli can be presented above the cognitive threshold, rather than self-related information below the cognitive threshold. The memory advantage of individual information provides theoretical support for the security of identity authentication. Self-cognition is the only result of long-term memory. It is difficult for intruders to forge the brain response of real users through short-term memory. The memory advantage is also reflected in functional magnetic resonance imaging. The memory function area located in the hippocampus is activated during self-recognition. The researchers considered whether visual stimuli could be presented to the participants in a subliminal manner. Since the stimulus is presented under the human perception threshold, the subjects will not be able to consciously change their processing of visual stimuli, thereby improving the security of the system.

Self-cognition uniqueness inspires individual identity feature extraction methods. In the feature extraction of identity time domains, with the advantage of high temporal resolution of EEG, attention should be give to the self-specific components P300, N250 and N170. In the process of identity authentication, the time domain is segmented according to the EEG components, and the corresponding weights are given to the characteristics of different stages according to the component differences of self and non-self signals. The identity features of each component fusion contribute to the improvement in classification performance. In the feature extraction of identity frequency domain, studies have found that self-uniqueness is more obvious in low frequency, and the low-frequency energy difference of 3–5 hz can be used as an auxiliary identity feature. The conclusion of important brain regions in self-specific research is helpful for spatial feature extraction, and the lead weight can be configured according to the important brain regions in self-identification. At the same time, the attention mechanism in the application neural network can be combined to automatically learn the spatial characteristics of identity, and the spatial identity information can be integrated by data-driven methods. Researchers have explored the application of nonlinear features and entropy features, such as sample entropy, information entropy, and Lyapunov that quantify the randomness of brain waves. The entropy features enrich the feature content, and the individual difference features contained in them are even better than the conventional methods. With the advantages of EEG dense electrode arrays, the network topology of brain activity is more applied. The brain network features include important nodes and edge connection weights. Common brain network construction methods include correlation, coherence, Granger causality, directed ADTF, etc. Brain networks are used to measure the interaction and connectivity between electrodes, and the salient features of brain connectivity have been applied to biometrics as extensions and supplements [93]. With the continuous development of graph neural networks in deep learning, GNN is continuously applied to brain network pattern recognition of individual identity features. In addition, according to the priority characteristics of self-information processing, the latency of EEG components can be used as the classification feature of identity authentication. For example, the experiment includes behavioral data such as buttons and eye movements, and the action response time can also be included in the identity feature.

## 6. Conclusions

This paper summarizes the uniqueness of self-information in behavioral, EEG components and fMRI imaging, comprehensively analyzes the multimodal brain imaging joint acquisition method, and discusses the neural mechanism of self-cognition from multiple perspectives. The uniqueness of self-information processing is manifested in faster responses, higher attention and stronger memory. The frontal lobe and parietal lobe are the most intense areas of specific responses. The amplitude and latency of P300 in EEG components are the most significant features. The above brain regions and signal components provide reference for data segmentation, lead selection and advanced feature construction. In addition, the wide application of self-information materials in the field of brainprint recognition reflects the important value of self-uniqueness in the field of biosafety detection.

## Figures and Tables

**Figure 1 biology-12-00486-f001:**
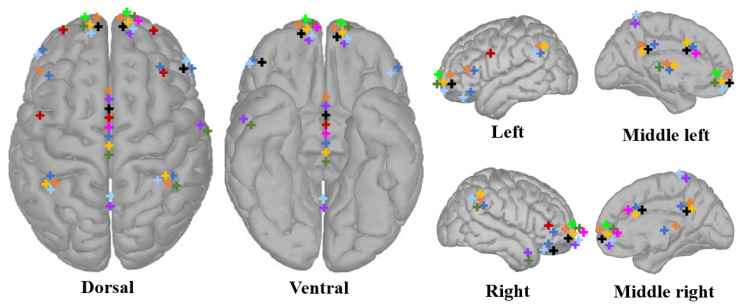
The cerebral cortex region of self-information processing.

**Figure 2 biology-12-00486-f002:**
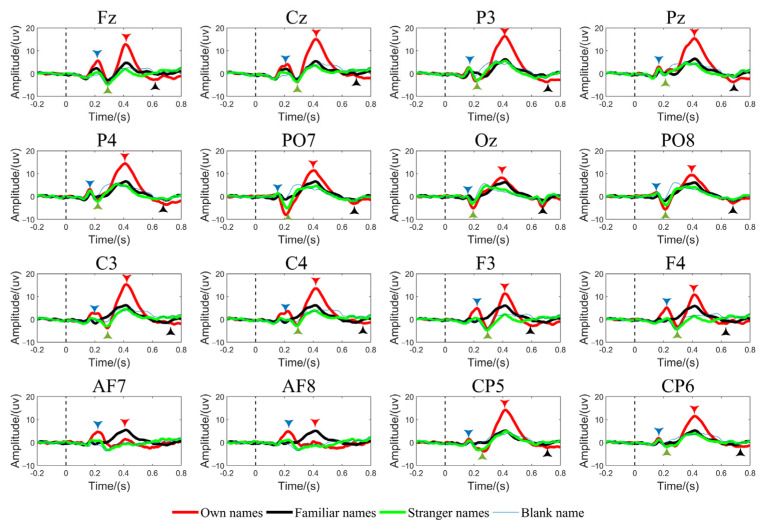
EEG components of self-information processing.

**Figure 3 biology-12-00486-f003:**
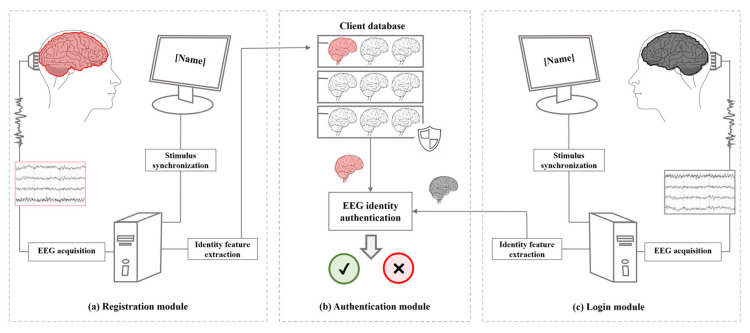
Flow chart of the brainprint recognition application.

**Table 1 biology-12-00486-t001:** Behavioral priority of self-information.

Behavior	Author	Experiment	Control Group	Conclusion
Keys	Tong [11]	Vision: face orientation recognition	Stranger face.	Own face stable faster than stranger.
	Keenan [35]	Vision: face recognition	Familiar colleagues, stranger face.	Upright and inverted faces, self-faces were significantly faster than familiars and stranger faces.
	Sui [36]	Vision: face reaction	Familiar, stranger face.	Recognition speed: self (551 ms), familiar (596 ms), unfamiliar face (588 ms).
	Jie [37]	Vision: face recognition	Familiar, disordered face.	Recognition speed: own face stable faster than stranger.Recognition accuracy: own face is higher than familiar and unfamiliar faces.
	Ma [38]	Vision: face orientation recognition	Familiars, disordered face.	The speed of self is faster than that of acquaintances and disordered faces, and it has the advantage of rapid recognition in both explicit and implicit experiments.
	Harris [39]	Vision: Name search	Interference name, ordinary words.	The names of 60 subjects were detected faster than other identities. The author believed that the subjects’ names were a ‘high-priority‘ text stimulus.
Eye track	Wang [40]	Vision: Name search	Mother, celebrity, common name.	The average number of saccades of one’s own name is 1.1 times less than that of a mother’s and a celebrity’s name, and the first fixation of one’s own name is 170 ms faster than that of other names.
EEG	Tacikowski [41]	Vision: name recognition	Celebrity, stranger name.	The P300 of one’s own name has a shorter latency period.

**Table 2 biology-12-00486-t002:** The sensitivity of self-information.

State	Author	Experiment	Control Group	Conclusion
Distraction	Moray [4]	Auditory: name recognition by the following ear	None	The brain can process self-names in a distracted state.
	Alexopoulos [23]	Auditory: name recognition	Acquaintance, stranger name	Self-attention is automatic, unintentional, unconscious, and uncontrolled.
	Pfister [24]	Vision: name recognition	Unrelated nouns, non-words	Self-names attract attention and were prioritized.
Subliminal	Shapiro [25]	Vision: name recognition	Other names, common nouns	Own name shows stronger anti-interference ability.
	Shelley [26]	Vision: Own name masked probability	Other symbols	In the masking task, the probability of own names was identified is higher.
		Vision: The interference target of its own names.	Other symbols	Self-information strongly affects the distribution of attention in both conscious and unconscious states.
	Pannese [27]	Vision: Face and gender matching.	Acquaintance, celebrity, stranger face	Self-information acquires special cognitive processing, and self-face benefits from the early processing of the brain.
	Sun [28]	Vision: Gender judgment of name	Celebrity, stranger name	The accuracy of gender judgment of one’s own name under low load and short delay conditions is significantly better than that of other names.
Interferent	S Brédart [29]	Vision: facial interference	Friend name	Self-face has stronger inhibition ability to target stimulus, and the interference effect of self-face is much stronger than that of friends.
	Devue [30]	Vision: facial interference	Friend, stranger face	The own face can temporarily attract attention within the focus of attention.
	Devue [31]	Vision: facial interference	Friend, stranger face	The reaction time of the own face interference is longer, and the own face is more difficult to distract attention.
	Yamada [32]	Vision: Name interference	Other’s name	The distortion of vision space caused by the own name leads to the spatial distribution deviation of attention.
	Wolford [34]	Vision: Name interference and decision	Same surname string, blank word.	The strong interference of one’s own name to the main task has a lower perception threshold.
Minimum consciousness	Ian [35]	Auditory: Name of sleep state	Classmate name	The extraordinary ability to remember the own name in sleep.
	Perrin [36]	Auditory: name of sleep state	Other’s name	Own names cause N2, P3 to appear more frequently.
	Kurtz [38]	Auditory: Patient’s names during recovery and post-anesthesia	Noise	The patient‘s reaction to their name is more obvious.
	Fishback [39]	Questionnaire: Cognitive tests of the elderly at different stages of AD	Time, place, count words.	With the development of dementia, the elderly finally forget their own name.
	Perrin [21]	Auditory: Name of sleep state	Other’s name	The brain’s cognitive response to one’s own name during sleep is similar to that during wakefulness.
	Pratt [40]	Auditory: Name of sleep state	Unrelated words	The feature of own names ERP show significant differences.
	Perrin	Auditory: the name of the minimally consciousness patient	Stranger name	The significant stimulus materials of own names have semantic processing.

## Data Availability

Not applicable.

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
