# Peer review of "Specific Neural Mechanisms of Self-Cognition and the Application of Brainprint Recognition"

_biology, 2023, doi:10.3390/biology12030486_

Round 1
Reviewer 1 Report
The authors presented a review on person identification using EEG. Here is my review:
1. Some claims in the paper need to be strenghtened using recent references.
2. My primary concern is the application of this technology, especially since there are much easier methods for biometrics such as fingerprint and iris recognition technologies.
3. The references appear to be very old. They need to be significantly updated.

Author Response
Original Manuscript ID: biology-2256870
Article Title: Specific Neural Mechanisms of Self-cognition and the Application of Brainprint Recognition
To: Editor
Re: Response to reviewers
Dear Editor,
Thank you for allowing a resubmission of our manuscript entitled, with an opportunity to address the reviewers’ comments. We are honored to have the opportunity to submit manuscripts in the area of Biology. We hope that our research can help the study of self-uniqueness and the progress of brainprint.
We thank the editor and reviewer for their thoughtful and careful review of our manuscript and hope that these changes will make the manuscript suitable for publication. Detailed revisions will be listed point-to-point in the appendix.
We are uploading our point-by-point response to the comments, an updated manuscript with yellow highlighting indicating change. If any additional changes are necessary, please let us know we will respond immediately.
Best regards,
RongKai Zhang.
Reviewer#1, Concern # 1: Some claims in the paper need to be strenghtened using recent references.
Author response:
We are sorry that the previous manuscript cited too much old literature. We one-sidedly consider that the classic references are reliable and time-tested, but we forget the support of the latest research on the manuscript 's point of view. We summarize the references in recent years to enrich the content of the manuscript, hoping that the addition of new references can help the manuscript understand and expand more exploration. Most of the new references are about self-cognition in the past 3 years.
Author action:
Addition text:
The research on self-cognition [1] has always been a hot field in psychology, cognition and philosophy. Self is the cognitive basis [2] of the individual to the world, a large number of researchers in-depth study of self-problem.
Besides, the internal self also has external objective embodiment, which includes self-semantic concept and cross-modal information[3]. For example, self-face [4] is the most intuitive self-related representation and an important symbol of personal identity and self-concept [5].
The advantage of self also extends to the body parts [7] such as hands and feet[8]. Researchers [9] have found that the important position of self is also reflected in the effect of self-advantage, which is manifested in the uniqueness of self-information processing. Studies [10] have found that self-prioritization is not only effective in vision and hearing, but also extends to different senses such as taste and touch.
The study found that individuals induced specific early and late EEG components [14] [15]in the process of self-recognition, and unique functional area activation [16] appeared in the cerebral cortex.
The study identified important brain functional areas such as frontal lobe [19], cortical mid-line structure [20] and temporal lobe [21] of self-processing, and extensively discussed the brain skewness of self-processing. However, a single brain imaging method is difficult to observe the details of the cognitive process, which requires a combination of the temporal resolution of EEG and the spatial resolution of fMRI.
A number of studies have shown that individuals pay more attention [23] to stimuli that contain self-information, and the brain has a specific response to self-information[24]. Individual advantages can be summarized as priority [25], sensitivity [21] and long-term memory [26].
Self-related stimuli are self-prioritized in the three attention systems of vigilance, orientation and executive control[27]. Individuals’ behavior and neural response to their own infor-mation are significantly faster than others’ information[28]. Researchers observed the speed advantage [29] of self in face and name experiments, and the observation data of behavioral keys and neural reactions verified the priority of self-information processing [30].
Researchers [31] found that the perception threshold of self-information was lower. Sub-liminal experiments [32] showed that self-information was difficult to produce’ attentional blink’ and self-related stimuli were more difficult to be masked.
Stimulating materials rely on rich self-experience to form a reliable memory structure [33]. Self-reference provides more memory pathways and memory traces for stimulating materi-als[34].
Addition recent references:
- Desebrock, C. Self-Prioritization in Motor Responses. PhD Thesis, University of Oxford, 2022.
- Maire, H.; Brochard, R.; Zagar, D. A Developmental Study of the Self-Prioritization Effect in Children between 6 and 10 Years of Age. Child development 2020, 91, 694–704.
- Honda, T.; Nakao, T. Interoception Modulates the Self-Prioritization Effect. 2021.
- Veldhuis, A. The Influence of Self-Relevant Information on Attention and Memory. PhD Thesis, Oxford Brookes University, 2019.
- Desebrock, C.; Spence, C. The Self-Prioritization Effect: Self-Referential Processing in Movement Highlights Modulation at Multiple Stages. Attention, Perception, & Psychophysics 2021, 83, 2656–2674.
- Ota, C.; Nakano, T. Self-Face Activates the Dopamine Reward Pathway without Awareness. Cerebral Cortex 2021, 31, 4420–4426.
- Orellana-Corrales, G.; Matschke, C.; Schäfer, S.; Wesslein, A.-K. Does an Experimentally Induced Self-Association Elicit Affective Self-Prioritisation? Quarterly Journal of Experimental Psychology 2022, 17470218221124928.
- Schäfer, S.; Wesslein, A.-K.; Spence, C.; Frings, C. When Self-Prioritization Crosses the Senses: Crossmodal Self-Prioritization Demonstrated between Vision and Touch. British Journal of Psychology 2021, 112, 573–584.
- Niu, G.; Yao, L.; Kong, F.; Luo, Y.; Duan, C.; Sun, X.; Zhou, Z. Behavioural and ERP Evidence of the Self-Advantage of Online Self-Relevant Information. Scientific Reports 2020, 10, 1–10.
- Deng, N.; Sun, Y.; Chen, X.; Li, W. How Does Self Name Influence the Neural Processing of Emotional Prosody? An ERP Study. PsyCh Journal 2022, 11, 30–42.
- Muñoz, F.; Rubianes, M.; Jiménez-Ortega, L.; Fondevila, S.; Hernández-Gutiérrez, D.; Sánchez-García, J.; Martínez-de-Quel, Ó.; Casado, P.; Martín-Loeches, M. Spatio-Temporal Brain Dynamics of Self-Identity: An EEG Source Analysis of the Current and Past Self. Brain Structure and Function 2022, 227, 2167–2179.
- Koch, C.; others What Is Consciousness. Nature 2018, 557, S8–S12.
- Leszkowicz, E.; Maio, G.R.; Linden, D.E.; Ihssen, N. Neural Coding of Human Values Is Underpinned by Brain Areas Representing the Core Self in the Cortical Midline Region. Social Neuroscience 2021, 16, 486–499.
- Martínez-Pérez, V.; Campoy, G.; Palmero, L.B.; Fuentes, L.J. Examining the Dorsolateral and Ventromedial Prefrontal Cortex Involvement in the Self-Attention Network: A Randomized, Sham-Controlled, Parallel Group, Double-Blind, and Multichannel HD-TDCS Study. Frontiers in Neuroscience 2020, 14, 683.
- Jublie, A.; Kumar, D. Early Capture of Attention by Self-Face: Investigation Using a Temporal Order Judgment Task. i-Perception 2021, 12, 20416695211032990.
- Smith, D.; Wolff, A.; Wolman, A.; Ignaszewski, J.; Northoff, G. Temporal Continuity of Self: Long Autocorrelation Windows Mediate Self-Specificity. NeuroImage 2022, 257, 119305.
- Caughey, S.; Falbén, J.K.; Tsamadi, D.; Persson, L.M.; Golubickis, M.; Neil Macrae, C. Self-Prioritization during Stimulus Processing Is Not Obligatory. Psychological Research 2021, 85, 503–508.
- McCrackin, S.D.; Lee, C.M.; Itier, R.J.; Fernandes, M.A. Meaningful Faces: Self-Relevance of Semantic Context in an Initial Social Encounter Improves Later Face Recognition. Psychonomic Bulletin & Review 2021, 28, 283–291.
- Sui, J.; Rotshtein, P. Self-Prioritization and the Attentional Systems. Current opinion in psychology 2019, 29, 148–152.
- Schäfer, S.; Frings, C. Understanding Self-Prioritisation: The Prioritisation of Self-Relevant Stimuli and Its Relation to the Individual Self-Esteem. Journal of Cognitive Psychology 2019, 31, 813–824.
- Golubickis, M.; Falbén, J.K.; Ho, N.S.; Sui, J.; Cunningham, W.A.; Macrae, C.N. Parts of Me: Identity-Relevance Moderates Self-Prioritization. Consciousness and cognition 2020, 77, 102848.
- Å»ochowska, A.; Jakuszyk, P.; Nowicka, M.M.; Nowicka, A. The Self and a Close-Other: Differences between Processing of Faces and Newly Acquired Information. Cerebral Cortex 2023, 33, 2183–2199.
- Falbén, J.K.; Golubickis, M.; Tamulaitis, S.; Caughey, S.; Tsamadi, D.; Persson, L.M.; Svensson, S.L.; Sahraie, A.; Macrae, C.N. Self-Relevance Enhances Evidence Gathering during Decision-Making. Acta Psychologica 2020, 209, 103122.
- Liu, S.; Jia, Y.; Zheng, S.; Feng, S.; Zhu, H.; Wang, R.; Jia, H. An Experimental Study of Subliminal Self-Face Processing in Depersonalization–Derealization Disorder. Brain Sciences 2022, 12, 1598.
- Svensson, S.L.; Golubickis, M.; Maclean, H.; Falbén, J.K.; Persson, L.M.; Tsamadi, D.; Caughey, S.; Sahraie, A.; Macrae, C.N. More or Less of Me and You: Self-Relevance Augments the Effects of Item Probability on Stimulus Prioritization. Psychological Research 2022, 86, 1145–1164.
- Schäfer, S.; Wentura, D.; Frings, C. Creating a Network of Importance: The Particular Effects of Self-Relevance on Stimulus Processing. Attention, Perception, & Psychophysics 2020, 82, 3750–3766.
Revision location: We updated the manuscript in 1. Introduction, page 1-2, lines 29-76.
Revision location: We updated the manuscript in 2. Behavioral uniqueness of self-awareness, page 2-3, lines 92-113.
Revision location: We updated the manuscript in References, page 14-16, lines 682-749.
Reviewer#1, Concern # 2: My primary concern is the application of this technology, especially since there are much easier methods for biometrics such as fingerprint and iris recognition technologies.
Author response:
Thanks for your professional questions, we are sorry that the lack of traditional biometric analysis has affected your reading experience. In the revised manuscript, we added the content of current identity security and traditional biometrics, and analyzed the characteristics of such external biometrics. Further, the research elaborates the need for biometric authentication at a higher security level. Brainprint as an innovative identity authentication model, and the potential advantages of ‘Brainprint’ as an internal identity authentication method.
Author action:
Addition text: With the advancement of sensors and biotechnology, the identity authentication of external biometrics has become the mainstream of daily life. However, in the face of more complex and severe identity security attacks, we should not only confirm the external biological characteristics, but also protect the user's subjective will in identity authentication. Traditional biometric authentication technology still has some security risks. In fingerprint identification, criminals may synthesize false fingerprints through materials such as gelatin, thus deceiving the authentication system. Besides, some criminals may force legitimate users to complete authentication by force. These security risks put forward higher requirements for biometric-based authentication. How to achieve in vivo detection and stress resistance has become the focus of current research in this field. Therefore, exploring the inherent biometric authentication technology has become an important research topic in the field of identity security. The subjective initiative of the brain matches the cognitive needs of the intrinsic biological characteristics, protects the subjective certification intention of the coerced person, and enriches the reset replacement function of the intrinsic certification content.
Revision location: We updated the manuscript in 5. Application of self-specificity in brainprint recognition, page 12, lines 555-567.
Reviewer#1, Concern # 3: The references appear to be very old. They need to be significantly updated.
Author response:
Thanks for your professional advice, we have added some new references, most of which are new research findings in the past 3 years, which can support the views of this manuscript.
Addition recent references:
- Desebrock, C. Self-Prioritization in Motor Responses. PhD Thesis, University of Oxford, 2022.
- Maire, H.; Brochard, R.; Zagar, D. A Developmental Study of the Self-Prioritization Effect in Children between 6 and 10 Years of Age. Child development 2020, 91, 694–704.
- Honda, T.; Nakao, T. Interoception Modulates the Self-Prioritization Effect. 2021.
- Veldhuis, A. The Influence of Self-Relevant Information on Attention and Memory. PhD Thesis, Oxford Brookes University, 2019.
- Desebrock, C.; Spence, C. The Self-Prioritization Effect: Self-Referential Processing in Movement Highlights Modulation at Multiple Stages. Attention, Perception, & Psychophysics 2021, 83, 2656–2674.
- Ota, C.; Nakano, T. Self-Face Activates the Dopamine Reward Pathway without Awareness. Cerebral Cortex 2021, 31, 4420–4426.
- Orellana-Corrales, G.; Matschke, C.; Schäfer, S.; Wesslein, A.-K. Does an Experimentally Induced Self-Association Elicit Affective Self-Prioritisation? Quarterly Journal of Experimental Psychology 2022, 17470218221124928.
- Schäfer, S.; Wesslein, A.-K.; Spence, C.; Frings, C. When Self-Prioritization Crosses the Senses: Crossmodal Self-Prioritization Demonstrated between Vision and Touch. British Journal of Psychology 2021, 112, 573–584.
- Niu, G.; Yao, L.; Kong, F.; Luo, Y.; Duan, C.; Sun, X.; Zhou, Z. Behavioural and ERP Evidence of the Self-Advantage of Online Self-Relevant Information. Scientific Reports 2020, 10, 1–10.
- Deng, N.; Sun, Y.; Chen, X.; Li, W. How Does Self Name Influence the Neural Processing of Emotional Prosody? An ERP Study. PsyCh Journal 2022, 11, 30–42.
- Muñoz, F.; Rubianes, M.; Jiménez-Ortega, L.; Fondevila, S.; Hernández-Gutiérrez, D.; Sánchez-García, J.; Martínez-de-Quel, Ó.; Casado, P.; Martín-Loeches, M. Spatio-Temporal Brain Dynamics of Self-Identity: An EEG Source Analysis of the Current and Past Self. Brain Structure and Function 2022, 227, 2167–2179.
- Koch, C.; others What Is Consciousness. Nature 2018, 557, S8–S12.
- Leszkowicz, E.; Maio, G.R.; Linden, D.E.; Ihssen, N. Neural Coding of Human Values Is Underpinned by Brain Areas Representing the Core Self in the Cortical Midline Region. Social Neuroscience 2021, 16, 486–499.
- Martínez-Pérez, V.; Campoy, G.; Palmero, L.B.; Fuentes, L.J. Examining the Dorsolateral and Ventromedial Prefrontal Cortex Involvement in the Self-Attention Network: A Randomized, Sham-Controlled, Parallel Group, Double-Blind, and Multichannel HD-TDCS Study. Frontiers in Neuroscience 2020, 14, 683.
- Jublie, A.; Kumar, D. Early Capture of Attention by Self-Face: Investigation Using a Temporal Order Judgment Task. i-Perception 2021, 12, 20416695211032990.
- Smith, D.; Wolff, A.; Wolman, A.; Ignaszewski, J.; Northoff, G. Temporal Continuity of Self: Long Autocorrelation Windows Mediate Self-Specificity. NeuroImage 2022, 257, 119305.
- Caughey, S.; Falbén, J.K.; Tsamadi, D.; Persson, L.M.; Golubickis, M.; Neil Macrae, C. Self-Prioritization during Stimulus Processing Is Not Obligatory. Psychological Research 2021, 85, 503–508.
- McCrackin, S.D.; Lee, C.M.; Itier, R.J.; Fernandes, M.A. Meaningful Faces: Self-Relevance of Semantic Context in an Initial Social Encounter Improves Later Face Recognition. Psychonomic Bulletin & Review 2021, 28, 283–291.
- Sui, J.; Rotshtein, P. Self-Prioritization and the Attentional Systems. Current opinion in psychology 2019, 29, 148–152.
- Schäfer, S.; Frings, C. Understanding Self-Prioritisation: The Prioritisation of Self-Relevant Stimuli and Its Relation to the Individual Self-Esteem. Journal of Cognitive Psychology 2019, 31, 813–824.
- Golubickis, M.; Falbén, J.K.; Ho, N.S.; Sui, J.; Cunningham, W.A.; Macrae, C.N. Parts of Me: Identity-Relevance Moderates Self-Prioritization. Consciousness and cognition 2020, 77, 102848.
- Å»ochowska, A.; Jakuszyk, P.; Nowicka, M.M.; Nowicka, A. The Self and a Close-Other: Differences between Processing of Faces and Newly Acquired Information. Cerebral Cortex 2023, 33, 2183–2199.
- Falbén, J.K.; Golubickis, M.; Tamulaitis, S.; Caughey, S.; Tsamadi, D.; Persson, L.M.; Svensson, S.L.; Sahraie, A.; Macrae, C.N. Self-Relevance Enhances Evidence Gathering during Decision-Making. Acta Psychologica 2020, 209, 103122.
- Liu, S.; Jia, Y.; Zheng, S.; Feng, S.; Zhu, H.; Wang, R.; Jia, H. An Experimental Study of Subliminal Self-Face Processing in Depersonalization–Derealization Disorder. Brain Sciences 2022, 12, 1598.
- Svensson, S.L.; Golubickis, M.; Maclean, H.; Falbén, J.K.; Persson, L.M.; Tsamadi, D.; Caughey, S.; Sahraie, A.; Macrae, C.N. More or Less of Me and You: Self-Relevance Augments the Effects of Item Probability on Stimulus Prioritization. Psychological Research 2022, 86, 1145–1164.
- Schäfer, S.; Wentura, D.; Frings, C. Creating a Network of Importance: The Particular Effects of Self-Relevance on Stimulus Processing. Attention, Perception, & Psychophysics 2020, 82, 3750–3766.
Revision location: We updated the manuscript in References, page 14-16, lines 683-750.

Reviewer 2 Report
The manuscript “Specific Neural Mechanisms of Self-cognition and the Application of Brainprint Recognition" by Drs. Rongkai Zhang et al aimed to review recent recent scientific and methodological achievements in the field of neurobiological research on the concept of "self", self-identification and related neurobiological and psychological phenomena. The authors touched on an extremely important and complex topic, and their manuscript will certainly evoke responses in the scientific community.
Having no objections on the merits, I nevertheless have some questions for the authors:
To what extent is the "Brainprint" concept described by the authors related to the "EEG Brainprint" methodology described in recent publications? Authors can use this publication to clarify this issue:
Min Wang, Jiankun Hu, Hussein A. Abbass, BrainPrint: EEG biometric identification based on analyzing brain connectivity graphs, Pattern Recognition V 105, 2020, 107381
Experimentally, using fMRI, a related question about voice recognition has been investigated and published:
Callan et al, Song and speech: brain regions involved with perception and covert production; Neuroimage, 2006 1;31(3):1327-42.
The authors rightly point out that the sensitivity of their own information is also widely used in daily life, medical care is used to detect the basic function of the brain. But it should be noted that this issue has been investigated and described in a number of publications in recent years. In particular, mention should be made of:
C. Koch, What is consciousness? Nature 557 (2018) 3–5
Tsytsarev V., Methodological aspects of studying the mechanisms of consciousness. Behav Brain Res. 2022 Feb 15;419:113684.
Minor criticism:
LINE 380:
The authors write:
“fMRI results showed that a wide range of bilateral networks, including frontal lobe, temporal lobe, limbic system and subcortical structure, were activated in the process of self-name recognition compared with names of celebrities and strangers” -
It is necessary to specify which subcortical structures.
The concept is interesting, the manuscript is organized well, and written clearly, I was impressed. I will be happy to recommend the manuscript for the publication after minor correction, suggested before.
Author Response
Original Manuscript ID: biology-2256870
Article Title: Specific Neural Mechanisms of Self-cognition and the Application of Brainprint Recognition
To: Editor
Re: Response to reviewers
Dear Editor,
Thank you for allowing a resubmission of our manuscript entitled, with an opportunity to address the reviewers’ comments. We are honored to have the opportunity to submit manuscripts in the area of Biology. We hope that our research can help the study of self-uniqueness and the progress of brainprint.
We thank the editor and reviewer for their thoughtful and careful review of our manuscript and hope that these changes will make the manuscript suitable for publication. Detailed revisions will be listed point-to-point in the appendix.
We are uploading our point-by-point response to the comments, an updated manuscript with yellow highlighting indicating change. If any additional changes are necessary, please let us know we will respond immediately.
Best regards,
RongKai Zhang.
Reviewer#2, Concern # 1: To what extent is the "Brainprint" concept described by the authors related to the "EEG Brainprint" methodology described in recent publications? Authors can use this publication to clarify this issue: Min Wang, Jiankun Hu, Hussein A. Abbass, BrainPrint: EEG biometric identification based on analyzing brain connectivity graphs, Pattern Recognition V 105, 2020, 10738.
Author response:
Thank you for your recommendation of the literature, helping me to clarify the concept and technology of "brainprint". The introduction of the "brainprint" in this literature is summarized in our manuscript, and the literature is also cited.
Author action:
Addition text: There are considerable individual differences in brain structure and advanced cognitive function, which lays a foundation for the study of brain biological indicators [93].
Addition text: EEG has more advantages than traditional biometric patterns in anti-spoofing attacks, privacy compliance and liveness detection [93].
Addition text: Brain networks are used to measure the interaction and connectivity between electrodes, and the salient features of brain connectivity have been applied to biometrics as extensions and supplements [93].
Addition reference: 93. Wang, M.; Hu, J.; Abbass, H.A. BrainPrint: EEG Biometric Identification Based on Analyzing Brain Connectivity Graphs. Pattern Recognition 2020, 105, 107381.
Revision location: We updated the manuscript in 5. Application of self-specificity in brainprint recognition, page 12, lines 574-575.
Revision location: We updated the manuscript in 5. Application of self-specificity in brainprint recognition, page 12, lines 586-587.
Revision location: We updated the manuscript in 5. Application of self-specificity in brainprint recognition, page 12-13, lines 656-658.
Revision location: We updated the manuscript in reference, page 18, lines 865-866.
Reviewer#2, Concern # 2: Experimentally, using fMRI, a related question about voice recognition has been investigated and published: Callan et al, Song and speech: brain regions involved with perception and covert production; Neuroimage, 2006 1;31(3):1327-42.
Author response:
Thank you for your professional recommendation. This article has helped us improve the section on voice and self-perception. This authoritative fMRI experiment is interesting, which discusses the covert production of self-information processing.
Author action:
Addition text: Callan [46] explored the brain response to passive sound stimulation in fMRI experiments and found that voice processing involves self-monitoring functional areas. The increased use of auditory-motor self-monitoring leads to different activation of auditory-motor processing related brain regions in singing and speech.
Addition reference: 46. Callan, D.E.; Tsytsarev, V.; Hanakawa, T.; Callan, A.M.; Katsuhara, M.; Fukuyama, H.; Turner, R. Song and Speech: Brain Regions Involved with Perception and Covert Production. Neuroimage 2006, 31, 1327–1342.
Revision location: We updated the manuscript in 2.2. Sensitivity of self-information, page 5, lines 187-190.
Revision location: We updated the manuscript in reference, page 16, lines 774-775.
Reviewer#2, Concern # 3: The authors rightly point out that the sensitivity of their own information is also widely used in daily life, medical care is used to detect the basic function of the brain. But it should be noted that this issue has been investigated and described in a number of publications in recent years. In particular, mention should be made of: C. Koch, What is consciousness? Nature 557 (2018) 3–5. Tsytsarev V., Methodological aspects of studying the mechanisms of consciousness. Behav Brain Res. 2022 Feb 15;419:113684.
Author response:
Thank you for your recommendation, these two high level articles broaden our understanding of consciousness. These two articles help this manuscript expand many vivid and interesting examples.
Author action:
Addition text: Accidents and infections during surgical anesthesia can cause severe brain injury, which may cause patients to lose language interaction for several years after surgery [19]. There is an urgent need for a reliable detection of individual consciousness impairment or disability in clinical practice, which is a serious challenge to ensure the patient 's postoperative experience life. The Glasgow Coma Scale is used for coma diagnosis [60], which evaluates brain recovery and deterioration and predicts recovery. However, the evaluation of the scale is difficult to apply to the real-time surgical environment, and its quantitative score depends on personal experience.
Addition text: The review [60] summarizes the important role of the intraparietal sulcus ( IPS ) of the supraparietal lobule ( SPL ), the attention system of the large area cortex and the frontal parietal cortex located at the junction of the parietal lobe, occipital lobe and temporal lobe, which is related to the neural correlation ( NCC ) of functional mapping and search consciousness.
Addition text: A wide range of cortical areas, including parietal, occipital and temporal areas, are posterior hot areas [19], which play an important role in people 's recognition and cognition of outside world. Even if small areas of the hot cortex are removed, it can lead to the loss of the entire conscious content, such as the inability to recognize faces, colors, and contours.
Addition reference: 19. Koch, C.; others What Is Consciousness. Nature 2018, 557, S8–S12.
Addition reference: 60. Tsytsarev, V. Methodological Aspects of Studying the Mechanisms of Consciousness. Behavioural Brain Research 2022, 419, 113684.
Revision location: We updated the manuscript in 2.2. Sensitivity of self-information, page 6-7, lines 261-268.
Revision location: We updated the manuscript in 3. fMRI and PET imaging of self-information processing, page 7, lines 370-374.
Revision location: We updated the manuscript in 3. fMRI and PET imaging of self-information processing, page 9, lines 405-408.
Revision location: We updated the manuscript in reference, page 15, lines 717.
Revision location: We updated the manuscript in reference, page 17, lines 800-801.
Reviewer#2, Concern # 4: LINE 380: The authors write:“fMRI results showed that a wide range of bilateral networks, including frontal lobe, temporal lobe, limbic system and subcortical structure, were activated in the process of self-name recognition compared with names of celebrities and strangers” It is necessary to specify which subcortical structures.
Author response:
We are sorry that the vague expression of subcortical structure in the manuscript hinders your reading experience. We add the details of subcortical structure in the revised version.
Author action:
Addition text: The authors analyzed the cortical regions of the thalamus, caudate and lentiform nucleus in the subcortical structure, which play an important role in self-name recognition and perform rough and fast information processing.
Revision location: We updated the manuscript in 3. fMRI and PET imaging of self-information processing, page 9, lines 414-416.
